# Place cells on a maze encode routes rather than destinations

Roddy M Grieves[1,2], Emma R Wood[2], Paul A Dudchenko[1,2]*

[1]School of Natural Sciences, University of Stirling, Stirling, United Kingdom; [2]Centre for Cognitive and Neural Systems, Edinburgh Medical School: Biomedical Sciences, University of Edinburgh, Edinburgh, United Kingdom

**Abstract** Hippocampal place cells fire at different rates when a rodent runs through a given location on its way to different destinations. However, it is unclear whether such firing represents the animal's intended destination or the execution of a specific trajectory. To distinguish between these possibilities, Lister Hooded rats (n = 8) were trained to navigate from a start box to three goal locations via four partially overlapping routes. Two of these led to the same goal location. Of the cells that fired on these two routes, 95.8% showed route-dependent firing (firing on only one route), whereas only two cells (4.2%) showed goal-dependent firing (firing similarly on both routes). In addition, route-dependent place cells over-represented the less discriminable routes, and place cells in general over-represented the start location. These results indicate that place cell firing on overlapping routes reflects the animal's route, not its goals, and that this firing may aid spatial discrimination.

## Introduction

A long-standing view of the hippocampus is that it contains a neural representation of space, a 'cognitive map' (*Tolman, 1948*), that encodes locations via the spatial receptive fields of place cells (*O'Keefe and Nadel, 1978*; *O'Keefe, 1999*). However, when a rat repeatedly traverses the same location on its way to different destinations, the place fields of hippocampal place cells are strongly modulated by where the animal is going or where it has come from (*Wood et al., 2000*; *Frank et al., 2000*; *Ferbinteanu and Shapiro, 2003*; *Bower et al., 2005*; *Ainge et al., 2007*; *Ji and Wilson, 2008*; *Pastalkova et al., 2008*; *Ferbinteanu et al., 2011*; *Allen et al., 2012*; *Catanese et al., 2014*; *Ito et al., 2015*). This suggests that place cells represent not just near instantaneous location (*Muller and Kubie, 1989*), but also aspects of the animal's goal directed behaviour.

In tasks where consistent differences in place cell firing are observed, rats are usually well-trained and execute rapid trajectories to goal locations. For example, in a study by *Ainge et al., 2007*, rats ran up the central stem of a double-Y maze to gather reward in one of four goal boxes. Place fields on the central stem and on adjacent portions of the maze often exhibited strong modulation depending on the goal box the animal was headed towards (see Figure 3 in *Ainge et al., 2007*). One interpretation of this 'splitter cell' pattern of firing (hereafter differential firing) is that it represents the animal's intended destination.

However, another interpretation is possible. To reach each goal location, the rat traversed a partially overlapping, but distinct route. Each route was repeated multiple times until the reward was moved to a different goal box. It is possible that the prospective differential firing observed in the overlapping portions of the routes did not reflect the intended goal location per se, but rather the position along one of four separate trajectories. In this view, differential firing reflects routes, as opposed to goals. This distinction between executing a series of responses and

*For correspondence:
p.a.dudchenko@stir.ac.uk

Competing interests: The authors declare that no competing interests exist.

**eLife digest** How does the brain represent the outside world? One way of answering this question is to study the brains of rats, because the basic plan of a rodent's brain is similar to that of other mammals, such as humans. For example, the brains of rodents and humans both contain a structure called the hippocampus, which plays important roles in navigation and spatial memory. Cells within the hippocampus called place cells support these processes by firing electrical impulses whenever the animal occupies a specific location.

When a rat runs along a corridor in a maze, its place cells often fire as it approaches a choice point. A given place cell will typically fire before the rat chooses a path leading towards one particular location, but not before choices that lead to other locations. The firing that occurs prior to the choice point is termed "prospective firing". However, it is not known whether the prospective firing of place cells represents the rat's final destination, or the specific route the animal takes to get there.

To address this question, Grieves et al. designed a maze in which two different paths from a starting corridor led to the same goal location. If place cells represent the goal location, they should fire whichever route the rat chooses. However, if they represent the specific path the rat takes to the goal, they should fire on one or the other route, but not both.

Grieves et al. found that almost all place cells with prospective activity in the starting corridor fired on a single route, as opposed to firing on both routes to the common goal. This suggests that the prospective firing in the hippocampus reflects the route the animal will take, rather than its intended destination. A future challenge will be to understand how the way the hippocampus codes routes interacts with brain circuits that code for intended goals, and how the activity of these circuits influences the animal's ability to navigate.

learning a goal location has a long tradition in spatial learning (*Tolman et al., 1946*; *McCutchan, 1947*; *Restle, 1957*).

To distinguish between goal and route accounts of differential firing, we designed a new apparatus in which different overlapping routes led to the same goal (*Figure 1A and B*). If differential firing on the overlapping sections of different trajectories reflects the animal's intended destination, then firing on the overlapping sections of the two routes leading to the same goal should be similar (but should differ from firing on the routes leading to other goals). In contrast, if differential firing reflects position along a specific route, the firing on these two routes should differ (*Figure 1D*). Our results support the latter interpretation. Place cells with fields on the overlapping portions of different routes leading to a common goal showed strong differential firing, and failed to show similar firing on different routes leading to the same goal.

## Results

Rats (n = 8) were trained on a task in which they travelled from a start box to one of three goals boxes (Left, Centre, Right) to obtain a food reinforcement (*Figure 1A,B*). Each path from the start box to a goal location was assigned a route number: Route 1 was the left-most path to Left Goal Box; Route 2 was the centre-left path to Centre Goal Box; Route 3 was the centre-right path to Centre Goal Box; and Route 4 was the right-most path to Right Goal Box (*Figure 1B*). Daily, uninterrupted sessions were comprised of four blocks of at least 11 trials. Within each block of trials, the reward remained in the same goal box, and then was switched to another goal box for the next block of trials. Thus, each of the four routes was associated with reward, according to a counterbalanced schedule across days (see *Figure 1E*). During a daily session the trial duration, the number of error trials before discovering which goal box was rewarded at the beginning of each block, the number of error trials after discovering the rewarded goal box during each block, and the trajectory chosen on each trial were recorded.

### Rats learned the win-stay task

Over the course of the first 10 sessions prior to surgery, the number of errors the animals made after the first correct trial in each block of trials decreased significantly over sessions ($F_{(9,99)} = 5.26$,

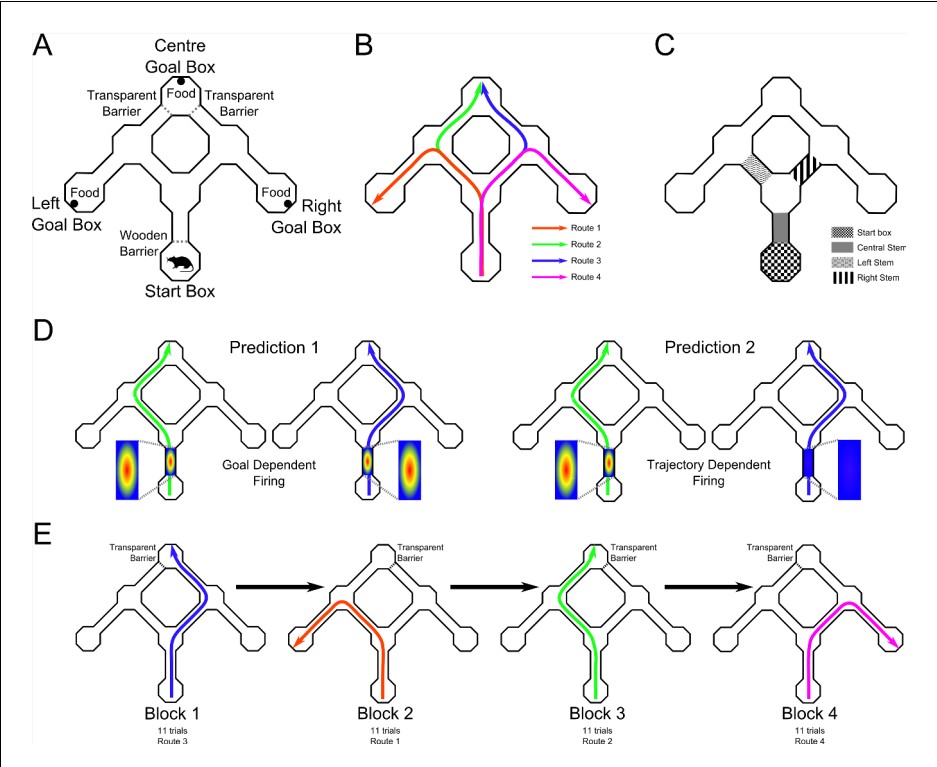

**Figure 1.** Maze apparatus with two routes leading to the same goal. (**A**) Top down view of the maze apparatus showing its layout including the start box, the three goal boxes, and the alleyways and choice points linking them. (**B**) The four trained routes through the maze. (**C**) Maze areas analysed for differential place cell firing. (**D**) Predictions of goal and route accounts of differential place cell firing. If differential firing of a place cells in the maze stem reflects the animal's intended goal (Prediction 1 - left plots), then a given cell should fire when the animals takes either the left or right route to the same goal. If such firing reflects the animal's route (Prediction 2 - right plots), firing should be seen on one route, but not the other. (**E**) Schematic of a representative daily session. Trials were blocked such that the same goal box was correct for at least 11 trials. The reward was then moved to a different goal box, and once it had been encountered by the rat, 11 further trials were run. In each session, all four routes were reinforced, although the order of these changed across sessions.

---

p<0.001, $\eta_p^2$ = 0.32; *Figure 2A* solid line). The time the rats took to complete each trial once they had located the rewarded goal in a block of trials also decreased significantly across sessions (*F*(9,99) = 6.87, p<0.001, $\eta_p^2$ = 0.38; *Figure 2B* solid line). In contrast, neither the number of errors per session (*F*(9,99) = 1.80, p>0.05), nor the time per trial (*F*(9,99) = 1.95, p>0.05) on trials prior to finding the rewarded goal box in each block changed significantly over sessions (*Figure 2A and B* respectively, dashed lines). Together, these results suggest that the rats could remember the last rewarded goal and learned to apply the win-stay, lose-shift rule, but that their efficiency in searching for the new goal location at the beginning of each block did not improve significantly across sessions.

Consistent with this, the rats navigated from the start box to a goal box significantly faster once they were aware of the reward location within each block of trials (*F*(1,11) = 25.83, p<0.001, $\eta_p^2$ = 0.70). Across all sessions, the rats averaged 9.08 s (S.D. = 6.19 s) to travel from the start box to the end of the maze on trials before they had identified the goal box which contained reward. Once the rewarded goal box had been visited, travel time on later trials in that block decreased to 5.38 s (S.D. = 3.48 s).

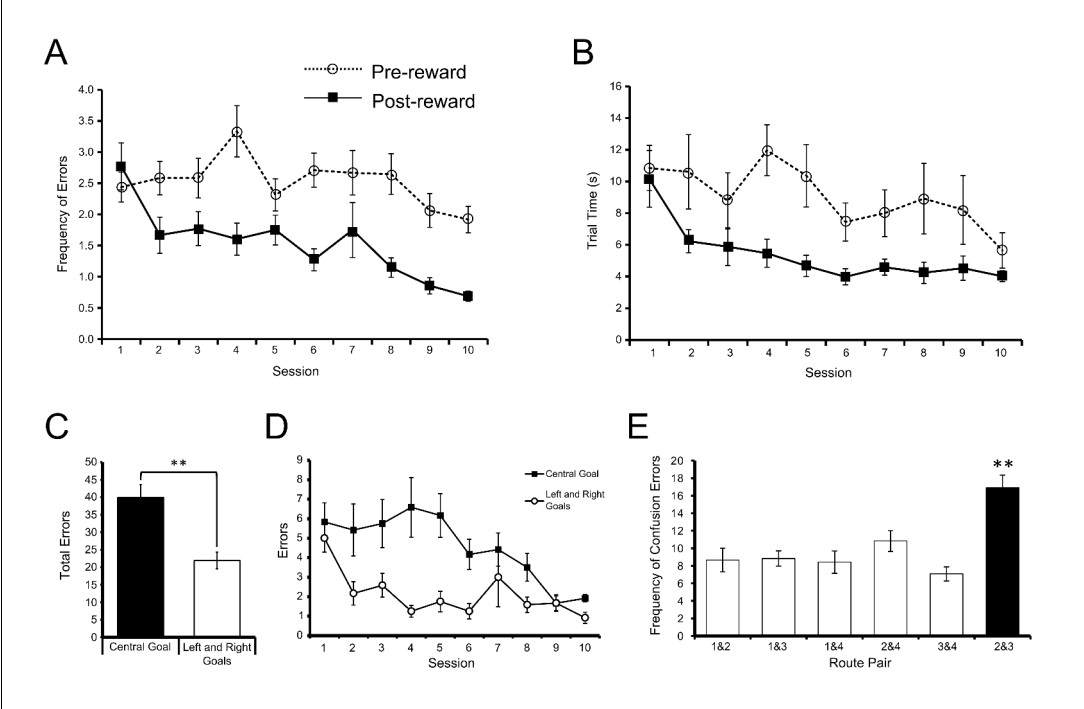

**Figure 2.** Acquisition of the win-stay task. (**A**) The mean number of errors preceding the identification of the reinforced goal box in each block of trials (broken line) did not change significantly across training sessions. However, the number of errors following the identification of the reinforced goal box in each block of trials (solid line) decreased significantly across training. (**B**) The mean time taken to complete each trial preceding the identification of the reinforced goal box (broken line) did not change significantly across training sessions. However, the time taken following the identification of the reinforced goal box (solid line) decreased significantly. Error bars depict SEM. (**C**) Mean total number of errors summed across 10 training sessions on trials in which the centre goal box was rewarded (black) and on trials in which the Left and Right Goal Boxes were rewarded (white), after the rewarded goal box had been identified in a block of trials. Rats made significantly more errors on trials when the two routes to the Centre Goal Box were rewarded than on trials when the two routes leading to the Left and Right Goal Boxes were rewarded. (**D**) Mean total number of errors on each session for trials on which the Centre Goal Box (black) and the Left and Right Goal Boxes (white) were rewarded. (**E**) Mean number of errors summed across 10 training sessions broken down by the nature of the error. For example, the first bar shows the average number of times the rats incorrectly chose Route 1 when Route 2 was rewarded, plus the number of times they chose Route 2 when Route 1 was rewarded. The number of confusion errors between routes to different goal boxes (hollow bars) was similar, regardless of route combination. However, there were significantly more confusion errors for the two routes to Centre Goal Box (filled bar). Error bars depict SEM.

## Routes to the same goal were more difficult to distinguish than routes to different goals

During training the animals made a greater number of errors on the trials when Routes 2 or 3 to the Centre Goal Box were rewarded than on trials when the outer routes (Routes 1 or 4) to the Left and Right Goal Boxes, respectively, were rewarded ($t(9) = 4.53$, $p<0.005$, paired t-test, see *Figure 2C*). This difference decreased across the training period (inner/outer goal x session interaction: $F(9,99) = 2.99$, $p<0.005$, $\eta_p^2 = 0.21$; *Figure 2D*).

We sought to define the nature of the errors which rats made after finding the location of the food reward in each block. An error where the rat took Route 1 to the Left Goal Box when Route 2 to the Centre Goal Box was rewarded can be interpreted as a similar form of navigation error as taking Route 2 to the Centre Goal Box when Route 1 to the Left Goal Box was rewarded. Both results may reflect an inability to discriminate between those two reward locations or routes. *Figure 2E* shows the distribution of post-reward errors when grouped into the six possible pairs of these confusion errors. From this figure it is clear that the rats made more errors between the two routes to the same goal (Routes 2 and 3 to the Centre Goal) as opposed to any other route pairs ($F(5,55) = 11.75$, $p<0.001$, $\eta_p^2 = 0.52$). Post-hoc multiple comparison tests confirmed Routes 2 and 3 were confused more than any other route pair ($p<0.05$ in all cases, with Sidak correction).

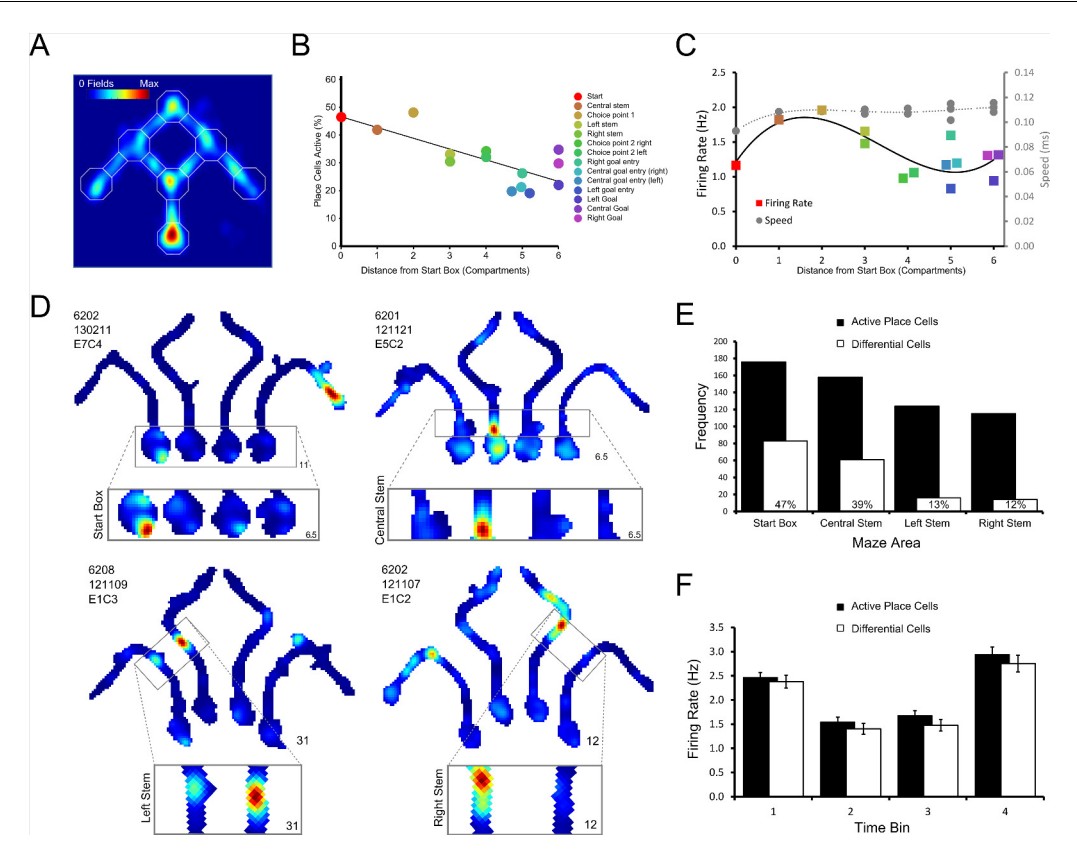

**Figure 3.** Differential firing throughout the maze. (A) Heat map showing the position of every place field recorded on the maze apparatus, the corresponding colour axis is shown in the top left corner. Place cells over-represented the start of the maze as shown by the large number of place fields observed there. (B) There was a linear decrease in the percentage of active (firing above 1Hz) place cells as the distance from the start box increased. (C) Place cell firing rates tend to be highest in the central stem and first choice point. This pattern approximately follows the average running speeds of the rats in the maze, except in the goal boxes where a slight increase in firing rate is also observed. Marker colours follow the key given for panel A. Lines show fitted 3rd order polynomials. (D) Representative example cells which, clockwise from the top left, show differential firing in the start box, central stem, left and right stems of the maze. (E) Number of place cells (black bars) active in each area of the maze tested for differential firing, and the number of these which are showed differential firing (hollow bars). The percentage of active place cells that showed differential firing in each area is indicated by the number written within the hollow bars. (F) The average firing rate of place cells and differential cells in the start box of the maze when this firing was divided into four bins of equal duration. Higher firing was observed in bin 1, just after the rat is placed in the start box and in bin 4, just before the holding barrier was removed.

The online version of this article includes the following figure supplement(s) for figure 3:

**Figure supplement 1.** Diagram of parameters used in all ANCOVA analyses.

## Single unit activity
### Place cells over-represent the start area of the maze

To test whether place cell activity encodes routes or goals, the trained rats were implanted with tetrodes targeting the CA1 cell layer of the hippocampus. In total, we recorded 377 place cells that were active on the maze from eight rats. We first analysed the distribution of place cell activity within the maze. The maze was divided into 14 sectors and place cells were categorised as being active in a given sector (defined as mean firing rate >1 Hz in that sector when the rat traversed one of the four trajectories) or not. Place cells were more likely to have a place field in the initial areas of the maze (e.g., the start box, central stem and first choice point) than in later ones (*Figure 3A*; see also *Ainge et al., 2007*). Consistent with this, there was a significant negative correlation between distance of the sector from the start box and the number of recorded place cells that were active in that sector ($r(12) = -0.78$, $p<0.05$; *Figure 3B*). Neither running speed, nor cell firing rate showed such a direct relationship with distance from the start box (*Figure 3C*).

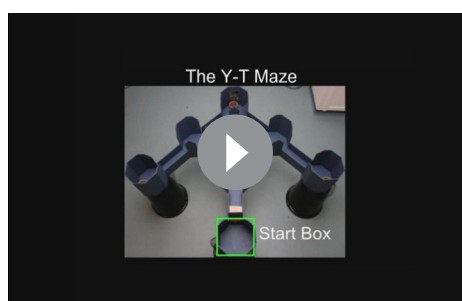

**Video 1.** Example of a place cell that fires differentially in the start box of the maze.
https://elifesciences.org/articles/15986#video1

## Differential place cell activity throughout the maze

As place field firing may be variable during different runs through a given location (**Fenton and Muller, 1998**), we assessed differential place cell activity using a non-parametric ranked analysis of covariance (rANCOVA, **Quade, 1967**). Additionally, both a permutation *F*-test and Generalized Linear Model provided similar results (see tables 1–4 in **Supplementary file 1**). Although the results of the rANCOVA are reported here, we note that other analyses may yield greater numbers of differential cells (**Prerau et al., 2014**). In the maze environment, 284 place cells were assessed by rANCOVA in at least one of the four sectors that were analysed for route/goal-dependent activity (**Figure 1C**; start box, central stem, left stem and right stem). Of these, 161 cells (57%) showed differential firing in at least one sector based on the rANCOVA results. Examining each of the four sectors of the maze separately, 83/176 assessed cells (47%) showed significantly different firing rates in the start box between the four trajectories (termed differential firing; see **Video 1**). 61/158 (43%) cells showed differential firing in the central stem, and 16/124 (13%) and 14/115 (12%) showed differential firing in the left and right stems respectively (**Figure 3E**). Examples of these cells can be seen in **Figure 3D**, **Figure 4** and **Figure 4—figure supplements 1–2**. In the start box, differential firing did not reflect anticipatory firing before the holding block was removed, as cells with differential firing did not differ from other active place cells (**Figure 3F**; ($F(1,257) = 0.25$, $p > 0.80$, $\eta_p^2 < 0.10$, tested by repeated measures ANOVA looking at firing rate x time bin (four equally spaced intervals) x cell type).

## Place cells encode routes, and not goals

The critical test of the route vs. goal account of differential activity is the comparison of firing rates on overlapping portions of the two routes (Routes 2 and 3) to the Centre Goal Box (**Figure 1D**). Therefore, we focussed only on the cells with differential activity, as identified by a rANCOVA analysis of firing rates in the start box or central stem. From post-hoc multiple comparison analyses between the four routes, we identified those cells that had significantly different firing on one of the four routes than on each of the other three (e.g. higher firing in the central stem on Route 1 than on Routes 2, 3 and 4), and also those cells which fired at a significantly different (higher or lower) rate on Routes 2 and 3 than on Routes 1 and 4, and whose firing rates did not differ between Routes 2 and 3. Examples of cells which exhibited differential firing are shown in **Figure 3D**, **Figure 4** and **Figure 4—figure supplement 1**. Overall, 48 cells were identified with one of these patterns of activity, and of these, 9 cells fired at a significantly different rate on Route 1 than on the other 3 routes, 13 cells fired at a significantly different rate on Route 2, 18 cells fired at a significantly different rate on Route 3, and 6 cells fired at a significantly different rate on Route 4. Only two cells fired similarly and at a significantly higher rate for Routes 2 and 3 than for Routes 1 and 4. The probability of observing this pattern by chance is low ($X^2(4, N = 48) = 15.96$, $p < 0.004$, Fischer's exact test, **Figure 5A**).

This pattern of firing reveals two interesting results. First, for the majority (95.8%) of place cells which fired significantly differently on at least one of the routes to the Centre Goal Box, firing appeared to be related to the specific intended route. In only two (4.2%) of the cells did the firing rate appear to be related to the intended goal, independent of the route. This pattern of results is significantly different than expected if equal numbers of cells had coded Route 2, Route 3 and both routes ($X^2(1, N = 33) = 25.49$, $p < 0.0001$, Fischer's exact test). Second, of the cells included in this analysis, 69% (33/48) fired preferentially on one (or both) routes to the Centre Goal Box (Routes 2 or 3), whereas only 31% (15/48) fired preferentially on one of the routes to the outer goals (Routes 1 or 4). The probability of observing this pattern of results by chance is also low ($X^2(1, N = 48) = 6.75$, $p < 0.015$, Fischer's exact test), and indicates significantly more route coding for the two routes to the Centre Goal Box than to the two routes to the outer goal boxes. In the segments of the maze

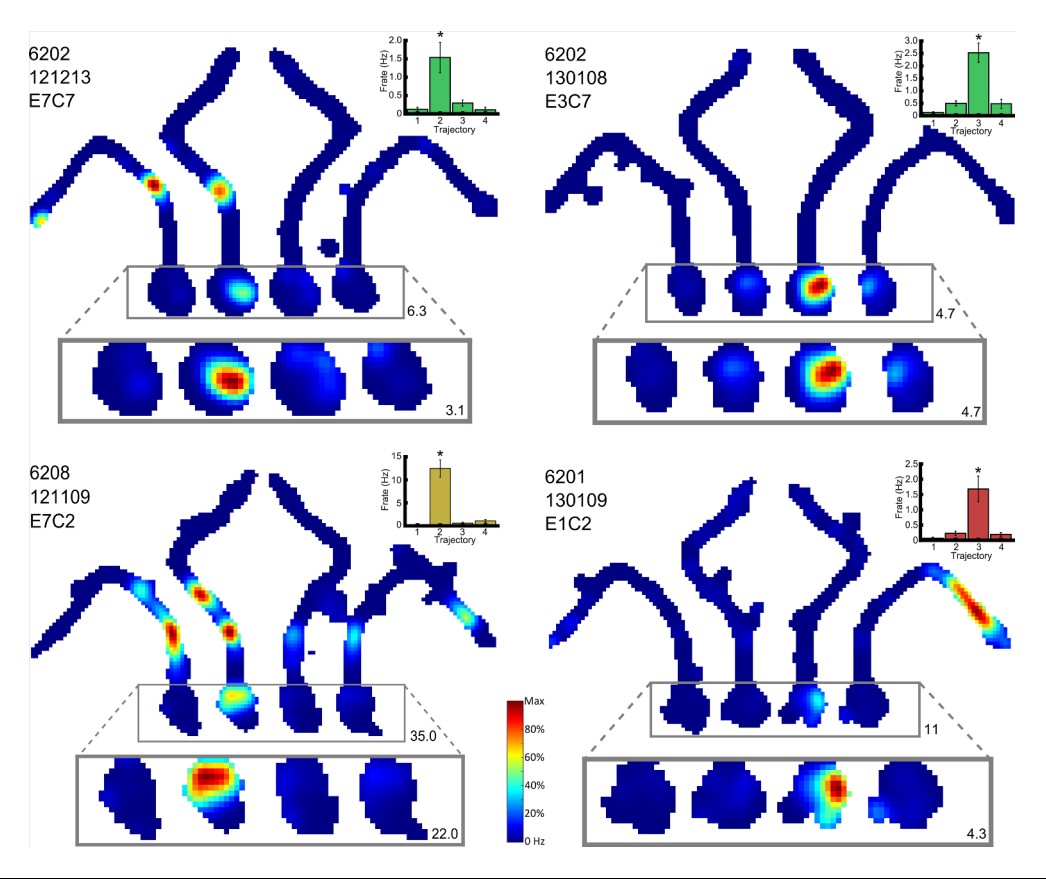

**Figure 4.** Four representative place cells, two per row, which show differential firing in the start box of the maze. For each cell, the firing rate map for the session is shown, but with data divided into trials in which the animals took each of the four possible routes. The area of this map in which differential activity was detected is highlighted, and this area is also shown, enlarged, below the firing rate map. The colour axis for the main and enlarged maps is scaled from 0Hz to the maximum firing rate in the map. The colour bar for these is given between the last two cells. The maximum firing rate in each map is denoted by a number found to its bottom right. The rat number, date of recording, electrode and cluster are given at the top left of the main firing rate map. The mean and SEM firing rate for the four trajectories is shown in a bar plot to the top right of the man firing rate map. These bars are coloured differently for each rat. Further examples can be found in *Figure 4—figure supplement 1*.

The online version of this article includes the following figure supplement(s) for figure 4:

**Figure supplement 1.** Representative differential cells included in the route/goal analysis.

**Figure supplement 2.** Four examples of differential firing on all trials within a session.

just after the first choice point, 30 cells (16 or 12.9% in the left stem, 14 or 12.2% in the right stem) were found to fire significantly differently depending on the trajectory of the animal. Over-representation of Routes 2 and 3 also appeared to be present here, however this effect is not statistically significant ($X^2(1, N = 30) = 0.53$, p>0.5, Fischer's exact test; *Figure 5B*).

## Cell isolation and differential firing

Place cells with significant differential activity on the maze did not differ from place cells without such activity in terms of isolation distances ($D = 0.10$, p>0.90), distribution of $L_{ratios}$ ($D = 0.25$, p>0.40), signal-to-noise ratios ($D = 0.2$, p>0.70), or peak waveform amplitude ($D = 0.2$, p>0.70) (*Figure 6B*). All tests were pairwise, Kolmogorov-Smirnov tests. Nor did we detect any relationship between these four measures and rANCOVA $F$-statistic ($r(573) = 0.05$, p>0.20, $r(573) = -0.08$, p>0.05, $r(573) = 0.03$, p>0.40 and $r(573) = 0.01$, p>0.05). All tests were Spearman's pairwise

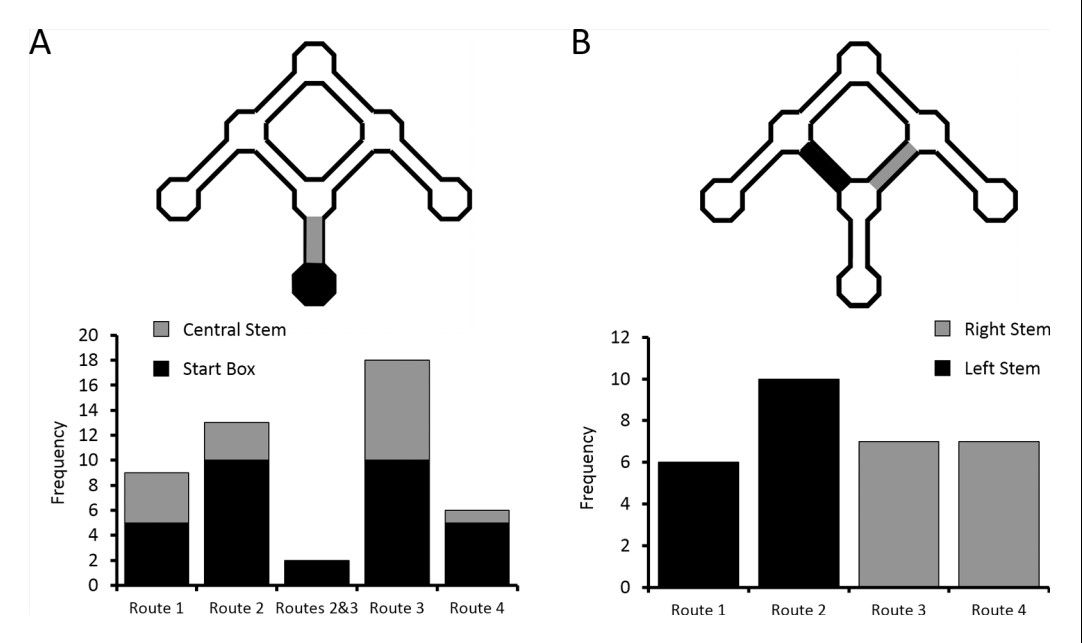

**Figure 5.** Distribution of place cells with differential firing. (**A**) Number of differential cells in the start box and central stem of the maze, sorted by preferred route. (**B**) Number of differential cells in left and right arms of the maze, again sorted by preferred route. See *Figure 3—figure supplement 1* for a breakdown of the parameters used by the ANCOVA analyses to determine differential activity. See *Supplementary file 1* (tables 1–4) for the results of the alternative differential activity statistical methods.

correlations, conducted after removing statistical outliers using an iterative implementation of the Grubbs Test (*Grubbs, 1969*) (see *Figure 6C*).

## Ensemble activity suggests route-dependent firing, not goal location-dependent firing

To test the accuracy with which an animal's trajectory could be identified based on the average firing rate of place cell ensembles we analysed 25 ensembles from six rats with an average of 11.12 cells/ensemble (SD = 5.67). Ensembles had to contain at least six place cells, and the largest ensemble contained 27 cells.

At the ensemble level, population vectors representing trajectories along Routes 1–4 in the start box and central stem were consistently matched to their correct goal population vector at a rate greater than incorrect goal population vectors (*Figure 7A* - start box, and *7E* - central stem). These matches were made at an above chance level, regardless of the final goal destination ($p < 10^{-3}$ in all cases, *Figure 7D and F* respectively). Furthermore, population vectors representing trajectories on the central two routes leading to the same goal were not matched to the alternative central route at an above chance level ($p > 0.9$ in all cases). These results confirm that at the ensemble level, as with the single cell level, place cell firing patterns were not dependent on the final goal destination but on the immediate trajectory. Furthermore, this information alone was sufficient to decode the animal's future destination at an above chance level. Also, there was no relationship between the position of a trial within a block and the likelihood of this trajectory being correctly matched to its goal population vector or the cosine similarity value resulting from this comparison ($p > 0.05$ in both cases) (*Figure 7—figure supplement 1*) suggesting that this trajectory coding persisted throughout blocks of trials.

## Place cells represent the Centre Goal Box similarly, regardless of the route taken there

It is possible that without having free, contiguous exploration of the maze rats considered the Centre Goal Box to be two distinct locations depending on the route taken. To test this possibility we analysed place cell firing rates in the Centre Goal Box and compared this firing between trials when

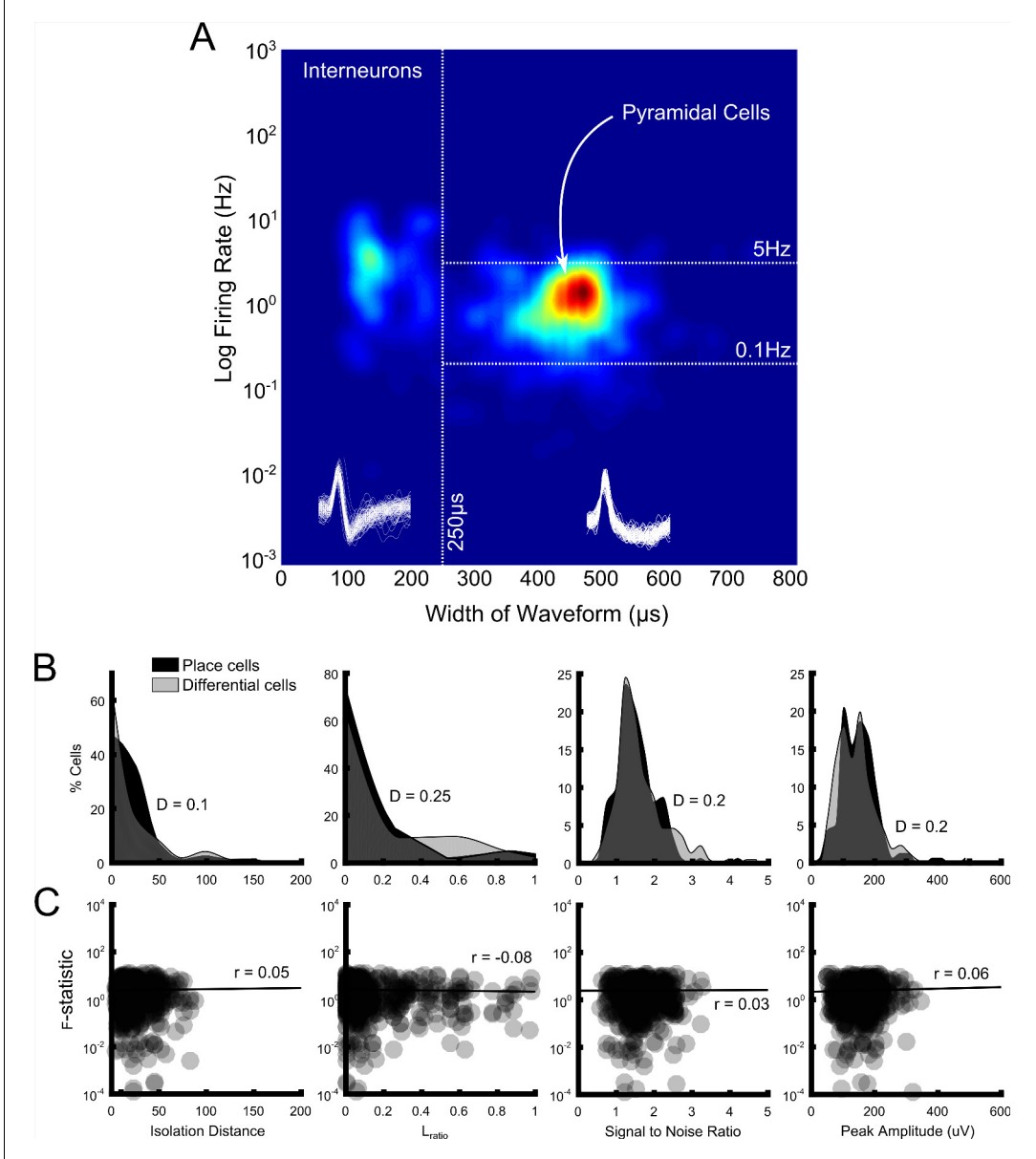

**Figure 6.** Population characteristics and signal quality measures and their impact on rANCOVA outcome. (**A**) Density plot showing firing rate plotted against width of waveform for every cluster (n = 641) in our population. Our cut-offs for characterising units as pyramidal cells are shown as white dotted lines. We also used a spatial information content threshold of 0.5b/s to further differentiate pyramidal cells from place cells. (**B**) Distribution of isolation distance (left), $L_{ratio}$ (middle left), signal to noise ratio (middle right) and peak amplitude (right) values for place cells that did not show differential activity (black shaded area) and for place cells which fired differentially on the maze (grey shaded area). These measures of cluster and signal quality did not differ significantly between the two groups of cells (p>0.05 in all cases, Kolmogorov-Smirnov tests, test statistics are shown on each plot). (**C**) The relationship between isolation distance (left), $L_{ratio}$ (middle left), signal to noise ratio (middle right) and peak amplitude (right) values and rANCOVA F-statistics calculated for all four maze areas assessed for differential firing. Each point represents a place cell assessed for differential firing in a maze segment (thus there can be a maximum of four points for one cell); darker areas denote overlapping points and thus data density. None of these measures are significantly correlated with rANCOVA outcome and thus differential firing (p>0.05 in all cases, Spearman's pairwise correlations, test statistics are shown on each plot).

rats navigated to the box using Routes 2 or 3 (*Figure 8A*). We found that firing in the Centre Goal Box when the rat took Routes 2 and 3 was correlated more highly (r(375) = 0.5, p<0.001) than firing between any other pair of goal boxes (r < 0.4 in all cases), such as the Left and Right Goal Box (*Figure 8B* left). We performed a shuffled analysis to determine if this correlation was greater than might be expected by chance. This analysis revealed that this correlation was in fact higher than

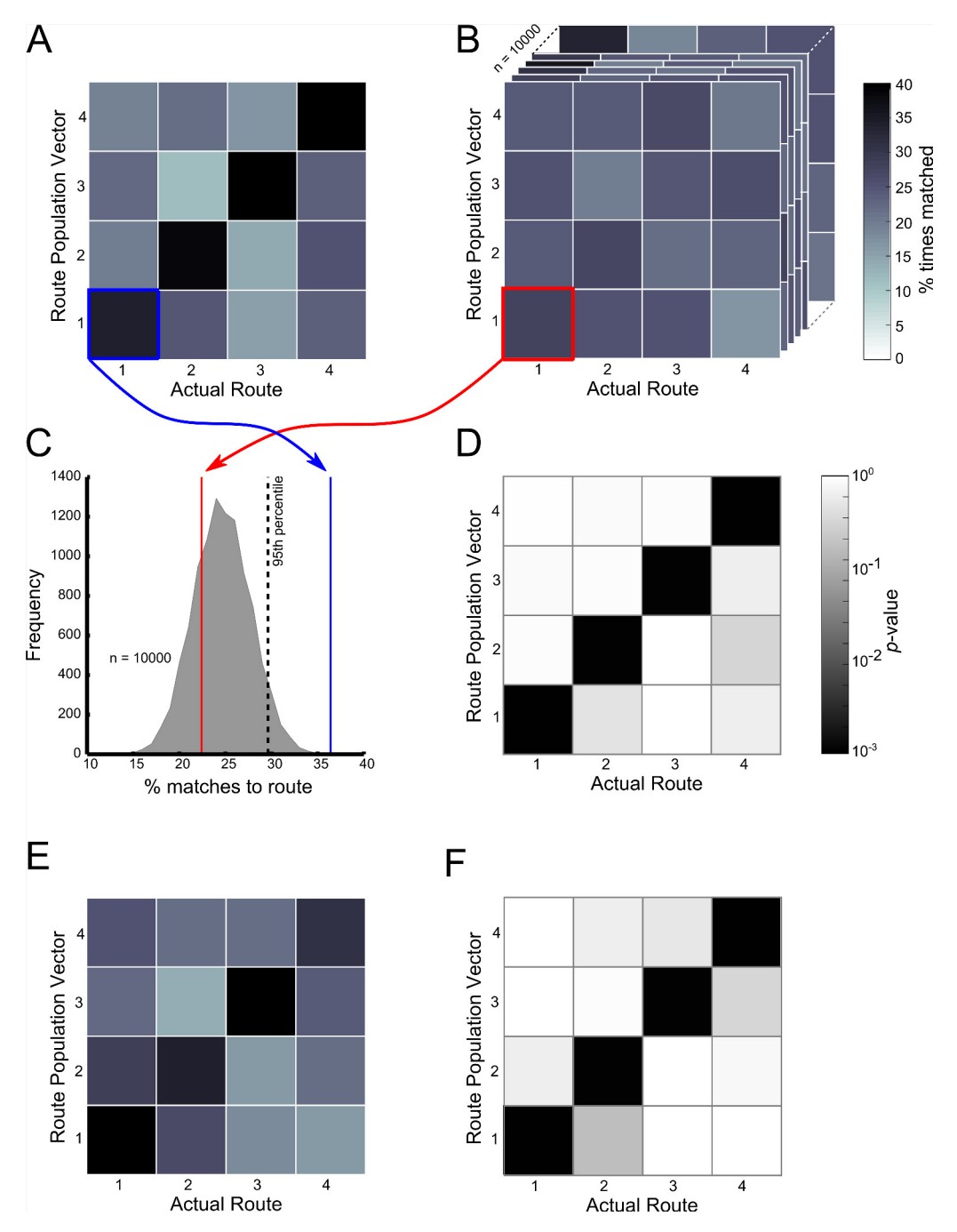

**Figure 7.** Ensemble decoding of trajectories (routes). The colour bar for plots (**A–C**) is given to the right of (**C**), the colour bar for (**E**) and (**F**) is given to the right of (**F**). (**B**) Trajectory population vectors (PVs) were compared to session PVs and matched according to highest cosine similarity score. Tiles here show the percentage number of each trajectory matched to each of the session PVs only for the start box of the maze. For example the tile highlighted in blue shows the percentage number of Route 1 PVs correctly matched to the session PV for Route 1. (**A**) Matches were also made using shuffled data, where the session PVs were randomly shuffled for each neuron. The tiles here show the same as (**B**) except that this data is for one shuffle (10000 were conducted in total). (**C**) Shows how the contents of the lower left tiles in (**A**) and (**B**) can be used to calculate a probability or p-value for the number of matches made here. The grey shaded curve shows the distribution of percentage correct matches for Route 1's PV to its session PV for all 10000 shuffles. The red line indicates where the value from the shown shuffled tile would fall. The blue line shows where the value from B (the actual data) would fall. A kernel smoothed cumulative density estimate (Epanechnikov) was then used to calculate the percentile value or probability of the real data value, given the shuffled distribution. (**E**) Shows the results of this analysis, with the probability of the percent correct matches made in (**B**), given the distributions in (**A**). Note that trajectories are only significantly matched to their corresponding session PVs. Furthermore, Routes 2 and 3 are

*Figure 7 continued on next page*

*Figure 7 continued*
not matched to each other significantly more than would be expected by chance. (C) and (F) show the same as (B) and (E) respectively, but for the central stem of the maze. See *Figure 7—figure supplement 1* for ensemble analysis at the single trial level.

The online version of this article includes the following figure supplement(s) for figure 7:

**Figure supplement 1.** Analysis of ensemble decoding within blocks of trials.

would be expected by chance (p<0.003, *Figure 8B* right) and that this was not the case for any other box pairs (p>0.09 in all cases). These results confirm that place cell activity in the Centre Goal Box was similar regardless of the route taken to get there, that this firing was more similar than in any other pair of goal boxes and that this firing was the only firing more similar than could be expected by chance.

## Histology

Electrode tracks confirmed placement of the electrodes in the CA1 region of the HPC (*Figure 9*). The final electrode placement for one of the eight animals was less obvious due to tissue damage near the implant site. For this animal, part of the electrode track was seen in the cortex above the HPC and appeared to have descended at the correct ML and AP coordinates, and complex spikes and theta oscillations were observed. The neurons recorded from this animal which passed our criteria for place cells in the maze apparatus did not differ from the other animals in terms of isolation distance, l-ratio, average firing rate, maximum firing rate (found in the firing rate map), width of waveform or spatial information content (p>0.05 in all cases, Exact Kolmogorov-Smirnov tests). Thus, this rat's data were included in the analyses above.

## Discussion

The current experiment tested whether the strong modulation of place cell firing that occurs when an animal travels to different destinations reflects the animal's route or the animal's intended destination. Our results, both at a single-unit and an ensemble level, suggest that such conditional firing encodes a learned route. We consider these findings below.

### Differential place cell firing

Previous studies have shown that when a well-trained rat runs through the common segment of a maze on its way to or from different destinations, clear modulation of hippocampal place cell firing rates is observed (e.g., *Wood et al., 2000*; *Frank et al., 2000*; *Ferbinteanu and Shapiro, 2003*; *Bower et al., 2005*; *Ji and Wilson, 2008*; *Pastalkova et al., 2008*; *Ferbinteanu et al., 2011*; *Catanese et al., 2014*; *Ito et al., 2015*). In many cases, however, it is unclear what this differential firing represents. In a continuous maze, differential firing appears to represent previous locations early in the common stem of the maze, and intended destinations later in the common stem (*Catanese et al., 2014*). In discrete trial tasks, where the animal is picked up and returned to a start box on each trial, the finding that place cells are active on journeys to one goal box and not active for journeys to other goal boxes has been interpreted as an encoding of the animal's intended destination (*Ainge et al., 2007*, *2012*).

However, an alternative explanation is also plausible. In the *Ainge et al. (2007)* experiment, as in other studies, animals were well-trained. The rats ran rapidly to the rewarded goal box with little hesitation along the route. Thus the modulation of place cell firing observed in this task might not represent the intended destination per se, but rather a read-out of a specific trajectory sequence (as in *Pastalkova et al. [2008]*). To differentiate between these possibilities, we tested animals on a task where two different routes led to the same goal location. Differential place cell firing on the overlapping portions of these routes would suggest that such activity encodes specific routes. Similar firing on the two routes would be consistent with the encoding of a common intended destination.

Our results demonstrate that differential firing is associated with the animal's route and not with its final destination. Of the cells with differential firing in the start box and central stem which fired preferentially on trajectories to the Centre Goal Box, nearly 96% showed differential firing between the two potential routes. Furthermore, at an ensemble level, the firing rate of place cells in the start box or central stem was sufficient to decode the animal's route at an above chance level, regardless

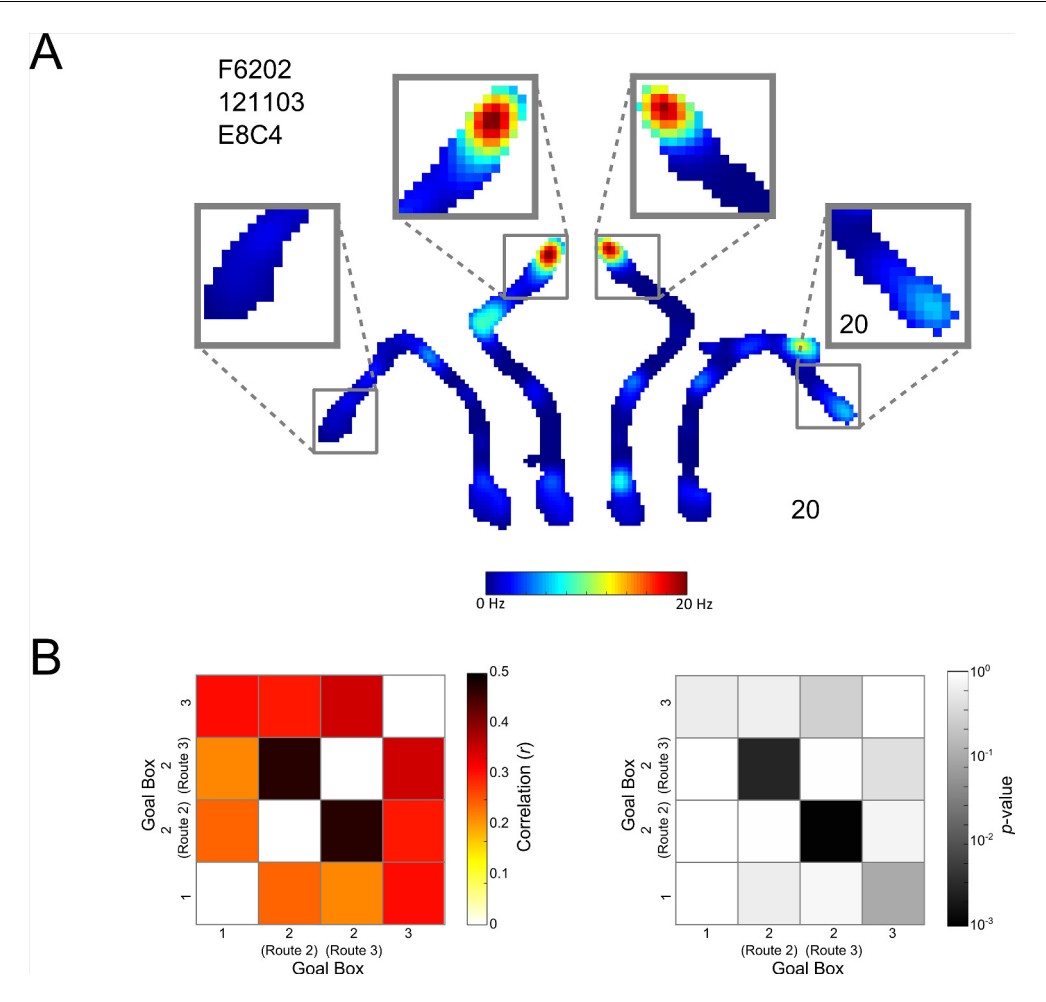

**Figure 8.** Place cell coding of the Centre Goal Box. (**A**) A representative cell which fires similarly in the Centre Goal Box regardless of which route the animal took to get there. The firing rate map for a whole maze session is shown. This is divided into the four possible trajectories but is plotted using one colour axis scaled from 0–20 Hz. Surrounding this, activity in each of the four goal boxes has been enlarged. This pattern of firing suggests that the animal was aware this goal box occupied a single spatial location and was thus one box at the end of two distinct trajectories. (**B**) Left, the result of Spearman's correlations comparing the firing of all place cells, from all rats and sessions in each pair of goal boxes, depending on the route taken to get there (i.e. firing in the Left Goal Box at the end of Route 1, the Centre Goal Box when at the end of Route 2, the Centre Goal Box at the end of Route 3 and finally the Right Goal Box at the end of Route 4). Correlations comparing the firing in each box to itself were not calculated and have thus been coloured white. Correlations are generally high, however, correlations comparing firing in the Centre Goal Box when it was accessed using the two different routes are the highest (i.e. Route 2 vs. Route 3 and vice versa), confirming that cells fire similarly in this box regardless of the route taken to get there. This suggests that rats and cells considered the Centre Goal Box to be one coherent spatial location. (**B**) Right, the results of a shuffling procedure to determine if the correlations are higher than would be expected by chance. Only those correlations between firing in the Centre Goal Box for Routes 2 and 3 are statistically significant (p<0.05). This test confirms that firing in the Centre Goal Box is more similar than would be expected by chance and that this firing is more similar than that between other pairs of goal boxes.

of which route was taken. This indicates that the ensemble firing of place cells carries sufficient information to distinguish the two central routes to the shared goal location and is thus also route-dependent, and not goal-dependent.

A potential caveat to these findings is that we may have biased the rats' representation towards routes by blocking one of the possible routes to the Centre Goal Box. We attempted to address this by training a second, naïve group of rats on the same task but without the transparent goal barrier.

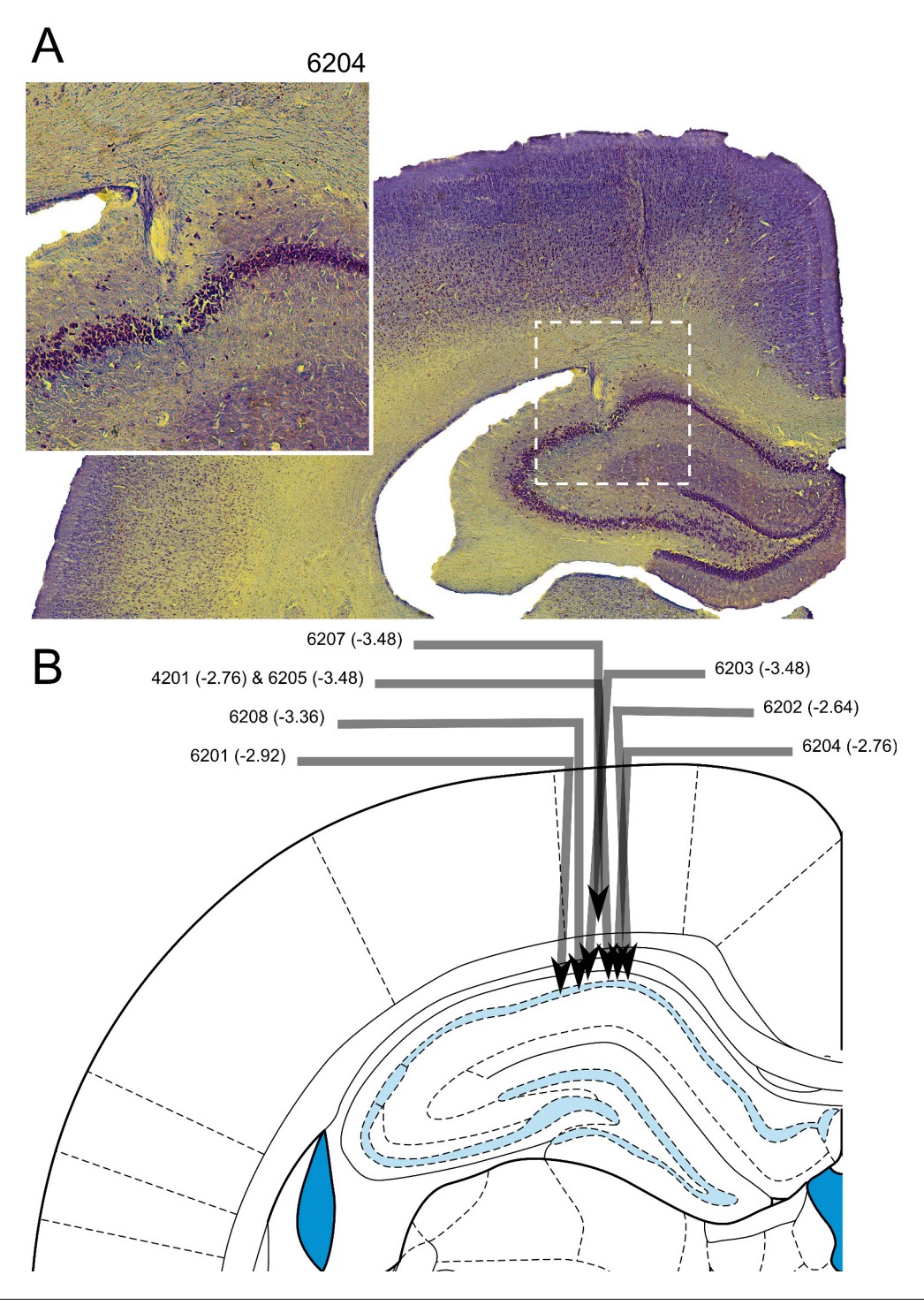

**Figure 9.** Histological confirmation of electrode placement. (**A**) Coronal section of hippocampus with electrode track (inset: higher magnification view). (**B**) Schematic of individual electrode tracks towards the CA1 cell layer of the hippocampus. Arrows represent the angle and depth of implantation with the arrow tip showing the point at which the electrode passed through the CA1 cell layer. No electrodes contacted lower cell layers. Each arrow is labelled with a rat number and an estimated anterior-posterior (AP) coordinate (the schematic shows a slice at an AP -3.48 mm from bregma - the intended coordinate).

However, the majority of these animals formed a strong bias for one of the two routes to the Centre

Goal Box. Thus, given the choice, the rats appeared to minimize the number of trajectories to learn, suggesting that even in the absence of a barrier, rats choose to solve the task in terms of routes, not goals. Similar navigation has been suggested in baboons, who simplify a complex jungle environment to a limited number of favourite routes (*Byrne et al., 2000*). The potential biasing of place cell representations is consistent with previous reports of differential place cell firing with respect to deprivation state (i.e., hunger or thirst) (*Kennedy and Shapiro, 2009*), spatial strategy (*Ferbinteanu et al., 2011*), type of reward (*Allen et al., 2012*), or order of non-spatial (olfactory) stimuli (*Allen et al., 2016*). Though route encoding is predominant in the current study, it is certainly possible that different task contingencies might yield a significant representation of individual goals.

Recent work by *Ito et al. (2015)* has suggested that trajectory-dependent cells are found in the nucleus reuniens, an input to CA1, and in the medial prefrontal cortex. They show that the nucleus reuniens appears necessary for the differential firing in CA1 place cells. Together with the current results, this finding implies that trajectory encoding may arise in the medial prefrontal cortex, and may be passed through the nucleus reuniens to the CA1 layer of the hippocampus.

The current results, like those of *Ito et al. (2015)*, deal only with prospective place cell firing. Findings from (*Ferbinteanu and Shapiro, 2003*) indicate that retrospective firing may not reflect trajectories. They trained rats on a plus maze task, and showed that even when the animal took indirect trajectories to a goal, differential retrospective firing was observed. That is, retrospective place cell firing depended on which maze arm the rat started from, regardless of its subsequent trajectory. A likely possibility is that the hippocampus represents both where the animal is going, and where it has been. Such an account is consistent with recent demonstrations of prospective firing (e.g., *Pfeiffer and Foster, 2013*), and with earlier lesion studies using retrospective homing tasks (*Gorny et al., 2002*; *Wallace and Whishaw, 2003*).

## Over-representation of trajectory starts

A second main finding in the current experiment was that the representation of the maze environment by hippocampal place fields was non-uniform. This occurred in two domains: the distribution of place fields on the maze, and the distribution between routes.

In the current experiment, we observed a linear decrease in the frequency of active place cells from the start box to the goal locations. In previous work, place cells appear to over-represent goal locations (*Markus et al., 1995*; *Hollup et al., 2001*; *Hölscher et al., 2003*; *Kobayashi et al., 2003*; *Hok et al., 2007*; *Dupret et al., 2010*). However, in previous studies with a double-Y or continuous-T maze, over-representation of the start areas of the maze has also been observed (*Ainge et al., 2007*, *2012*). One possibility is that in mazes where rats run overlapping routes but represent these independently, over-representation is expected for the common segments of the maze. A second, intriguing possibility is that the dorsal hippocampus represents distance to a goal, and this is more apparent in structured tasks with constrained routes. Recent findings suggest that in humans, the posterior hippocampus - which corresponds to the rodent dorsal hippocampus - exhibits more activation the farther one is from a navigational goal (*Howard et al., 2014*).

The second type of over-representation observed was a larger number of route-dependent cells coding specifically for the two routes to the Centre Goal Box than for routes to the two outer goal boxes. In learning the task, rats made significantly more errors when they were navigating these routes and they confused these two routes more than any other route pair. Over-representation may be the result of the recruitment of additional neural resources in the face of a difficult discrimination. The hippocampus has been implicated in the discrimination of structurally similar spatial environments (*Sanderson et al., 2006*; *Aggleton and Pearce, 2001*), and it is possible that the similarity of two central routes requires greater hippocampal resources to discriminate, yielding an increase in route specific firing.

## Summary

The current study makes three contributions. First, we show that the strong modulation of place fields when rats run through a common location on their way to different destinations reflects the encoding a specific route or trajectory, and not the encoding of an intended goal per se. Second, we show that routes leading to the same goal are more difficult to discriminate than routes leading to different goals. Finally, we found that place fields over-represent the early portions of the maze,

and difficult-to-discriminate routes. These results suggest that although the hippocampus represents places, the firing of place cells can also represent well-learned routes. It is possible that this representation, coupled with increased activity further from a goal, allow the animal to determine the distance and the route to a goal.

## Materials and methods

### Animals

For the behavioural portion of the experiment 12 male Lister hooded rats, with an average weight of 300 g, were used as subjects. Eight of these animals were subsequently used in the electrophysiological portion of the experiment at which point they weighed approximately 400–450 g. A further eight naïve animals, with an average weight of 300 g, were used to test an alternative training protocol. All animals were housed in groups of four in standard cages, but housed individually in custom designed cages after surgery. The animals were maintained under a 12 hr light/dark cycle and testing was performed during the light phase of this cycle. Throughout testing, rats were food restricted such that they maintained approximately 90% (and not less than 80%) of their free-feeding weight. This experiment complied with the national [Animals (Scientific Procedures) Act, 1986, United Kingdom] and international [European Communities Council Directive of November 24, 1986 (86/609/EEC)] legislation governing the maintenance of laboratory animals and their use in scientific experiments. Local ethical approval was granted by the University of Edinburgh Animal Welfare and Ethical Review Board.

### Electrodes and surgery

Microdrives were based on a modified tripod design described previously (*Kubie, 1984*). The drives were comprised of eight tetrodes, each of which was composed of four HML coated, 17 µm, 90% platinum 10% iridium wires (California Fine Wire, Grover Beach, CA). Tetrodes were threaded through a thin-walled stainless steel cannula (23 Gauge Hypodermic Tube, Small Parts Inc, Miramar, FL). The day before surgery and again immediately before surgery the tip of every electrode was gold plated (Non-Cyanide Gold Plating Solution, Neuralynx, MT) in order to reduce the impedance of the wire from a resting impedance of 0.7–0.9 MΩ to a plated impedance in the range of 200–300 kΩ (250 kΩ being the target impedance).

Electrodes were implanted using standard stereotaxic procedures under isoflurane anaesthesia. Hydration was maintained by subcutaneous administration of 2.5 ml 5% glucose and 1 ml 0.9% saline. Animals were also given an anti-inflammatory analgesia (small animal Carprofen/Rimadyl, Pfizer Ltd., UK) subcutaneously. Electrodes were lowered to just above the CA1 cell layer of the hippocampus (-3.5 mm AP from bregma, $\pm$2.4 mm ML from the midline, $-1.7$ mm DV from dura surface). The drive assembly was anchored to the skull screws and bone surface using dental cement. Animals were given at least two hours recovery in their home cage, heated to body temperature. Following this, at least one week of recovery time passed before animals were screened for cells. During this week, the animals' food was tapered from free feeding to the pre-surgery level of restriction.

### Unit recording

Single unit activity was observed and recorded using a 32-channel Axona USB system (Axona Ltd., St. Albans, UK). Mill-Max connectors built into the rat's microdrive were attached to the recording system via two unity gain buffer amplifiers and a light, flexible, elasticated recording cable. The recording cable passed signals through a ceiling mounted slip-ring commutator (Dragonfly Research and Development Inc., Ridgeley, West Virginia) to a pre-amplifier where they were amplified 1000 times. The signal was then passed to a system unit; for single unit recording the signal was band-pass (Butterworth) filtered between 300 and 7000 Hz. Signals were digitized at 48 kHz and could be further amplified 10–40 times at the experimenter's discretion. The position of the animal was recorded using infra-red LEDs fixed to the unity gain amplifiers attached to the rat's microdrive. A ceiling mounted, infrared sensitive CCTV camera tracked the animal's position. Rats were screened for single unit activity and for the presence of theta oscillations once or twice a day, five days a week.

## Apparatus

The maze environment was constructed from wood and consisted of seven octagonal enclosures (25 × 25 cm with 25 cm high walls), which formed the start box, the three choice points, and the three goal boxes (*Figure 1A*). Seven wooden alleyways connected these enclosures; these were 20 cm long × 10 cm wide with 10 cm high walls. All alleyways and octagonal enclosures were painted blue and the maze was elevated 60 cm from the floor on wooden stools. A moveable wooden barrier (10 cm wide 25 cm tall) could be placed at the exit of the start box to confine the rat to the start box. A moveable transparent Perspex barrier (also 10 × 25 cm) could be used at one or other entrance to the central goal box to prevent the animal entering the goal box via that entrance. The maze was curtained off from the remainder of the room on the left with a large white sheet. On the right wall there was a large window blackout shutter, and on the wall in front of the maze was an upwardly directed light source. To add to the distinctiveness of the goal boxes within the maze, each contained a different object: a small grey elephant statue, a small white opaque bottle with cork stopper, and a small black and red box with slanted lid. Each goal box also had a Latin alphabet character in a different reflective colour affixed to the wall. Heavy ceramic reward dishes were placed in each goal box, directly beneath these reflective letters. The start box did not contain any objects, but an orange fluorescent star was affixed to the block which kept the animal from entering the alleyways.

The square open field environment (100 × 100 cm with 25 cm high walls) was also constructed of wood and painted black. When in use (for screening for place cells), this box was placed on top of the maze apparatus, and so was elevated 85 cm above the floor, and was surrounded by the same set of distal cues.

## Win-stay task

The current task was similar to that employed in previous experiments using a double-Y maze (*Ainge et al., 2007*). A trial started when the experimenter raised the wooden block holding the animal in the start box at the base of the maze. The animal then navigated through the maze to one of the three goal boxes. On any given trial, only one goal box contained reward. Rats were not permitted to return towards the start box during a trial. If the rat entered the rewarded goal box, a correct choice was scored, and it was allowed to eat the food reward (CocoPops, Kelloggs, Warrington, UK) for a minimum of three seconds. The rat was then lifted by the experimenter, placed back into the start box and allowed to finish consuming any carried food reward. If the rat entered an unrewarded goal box, an incorrect choice was scored, and the animal was returned to the start box and held there using the wooden barrier for a minimum of three seconds.

For the Centre Goal Box, one of the two entrances was blocked with a piece of clear Perspex positioned in the doorway. Thus, on trials in which the Centre Goal Box was baited, only one of the two paths from the start box to this goal box (Routes 2 or Route 3) allowed access to the reward. For trials where the Left or Right Goal Boxes were reinforced, the transparent barrier was present, but was placed on the entry to the Centre Goal Box furthest from the rewarded box (i.e., at the right entry to the Centre Box if the Left Goal Box was reinforced). This was to ensure that the choice at the final junction was between two open goal boxes.

Rats completed trials as described above until they entered the correct goal box. They were then given a further 11 trials, and in these trials the same goal box was reinforced. At the end of this first block of trials, the food was moved to a new goal box and the process was repeated (*Figure 1E*).

## Win-stay with free choice

We trained an additional eight naïve rats on the same task described above in the absence of the Perspex barrier to the middle goal box. In this case, either route to the Centre Goal Box allowed access to the reward. As before, reward was available in one goal box for a block of trials, and the Centre Goal Box was reinforced for two blocks. We hoped that the rats would sample both routes at a roughly equal frequency without needing to direct their behaviour within blocks of trials to the Centre Goal Box. These rats were trained for a total of 12 sessions. Unfortunately, all of the rats rapidly developed a preference for only one route to Centre Goal Box (mean 71% of trials via preferred route), and thus this variant of the task was unsuitable for assessing differences in place cell activity between routes and goals.

## Place cell recording

After recovery from surgery, rats were screened daily for place cells in the open field apparatus. Upon the identification of place cells, an uninterrupted recording session was conducted. After a 10–15 min long recording session in the open field, rats were placed in the start box of the maze and the open field environment was removed. Rats were allowed a 60 s rest period in the start box before starting the win-stay, lose-shift task in the maze. The behavioural protocol during the maze phase was comparable to that used during pre-surgery training described above, with the exception that rats were required to make at least six correct trials within a block of trials before the reward location was changed. This was necessary to ensure adequate sampling of each trajectory. Between each trial, rats were confined to the start box for a minimum of six seconds (session mean = 9.34 s, SD = 0.62 s) in order to capture any possible differential firing which occurred there.

At the end of the recording session, the animals were removed from the maze apparatus and the electrodes were lowered in order to maximise the chance of recording from a different population of cells on the following day. No attempt was made to track cells across days, and thus a subset of the cells may have been recorded on more than one session. Rats were tested until cells were no longer observed (range 3–17 sessions).

## Cluster cutting

Single unit activity was analysed offline using a Matlab script that allowed the data files to be processed by the Klustakwik spike sorting program (*Kadir et al., 2013*). The dimensionality of the waveform information was reduced to first principal component, energy, peak amplitude, peak time, and width of waveform. The energy of a signal *x* was defined as the sum of squared moduli given by the formula:

$$\varepsilon_x \triangleq \sum_{n=0}^{N-1} |x_n|^2$$

Based on these parameters, Klustakwik spike sorting algorithms were then used to distinguish and isolate separate clusters. The clusters were then further checked and refined manually using the manual cluster cutting GUI, Klusters (*Hazan et al., 2006*). As well as the previously mentioned features, manual cluster cutting also made use of spike auto- and cross-correlograms. Cluster quality was operationalised by calculating isolation distance (Iso-D), $L_{ratio}$, signal to noise ratio (S/N) and peak waveform amplitude, taken as the highest amplitude reached by the four mean cluster waveforms. For cluster *C*, containing $n_c$ spikes, Iso-D is defined as the squared Mahalanobis distance of the $n_c$-*th* closest non-*c* spike to the centre of *C*. The squared Mahalanobis distance was calculated as:

$$D_{i,C}^2 = (x_i - \mu_C)^T \sum_C^{-1} (x_i - \mu_C)$$

where $x_i$ is the vector containing features for spike *i*, and $\mu_c$ is the mean feature vector for cluster *C*. A higher value indicates better isolation from non-cluster spikes (*Schmitzer-Torbert et al., 2005*). The *L* quantity was defined as:

$$L(c) = \sum_{i \notin C} 1 - CDF_{x_{df}^2} (D_{i,C}^2)$$

where $i \notin C$ is the set of spikes which are not members of the cluster and $CDF_{x_{df}^2}$ is the cumulative distribution function of the distribution with 8 degrees of freedom. The cluster quality measure, $L_{ratio}$ was thus defined as *L* divided by the total number of spikes in the cluster (*Schmitzer-Torbert and Redish, 2004*). As the signal and noise are both measured across the same impedance signal to noise ratio (S/N) was defined as:

$$Signal\ to\ Noise\ Ratio = \left( \frac{\left( \sqrt{\mu}_{signal} \right)^2}{\left( \sqrt{\mu}_{noise} \right)^2} \right)^2$$

where μ is the mean of the waveform amplitude. For noise we used the noise cluster which accompanied unit spikes on that tetrode. All four quality measures were assessed for their potential impact on our analyses by comparing the values observed in differential cells to non-differential place cells and by assessing the relationship between these measures and rANCOVA *F*-statistic (as suggested by *Schmitzer-Torbert et al., 2005*).

## Place cell identification

A cluster was classified as a place cell on the maze if it satisfied the following criteria: i) the width of the waveform was >250 μs, ii) the mean firing rate on the maze was greater than 0.1 Hz but less than 5Hz (see *Figure 6A*) and iii) the spatial information content was greater than 0.5 b/s. Spatial information content is given by the equation:

$$Information\ content = \sum P_i(R_i/R)\log_2(R_i/R)$$

where *i* is the bin number, $P_i$ is the probability for occupancy of bin *i*, $R_i$ is the mean firing rate for bin *i*, and *R* is the overall average firing rate (*Skaggs et al., 1993*).

Firing rate maps were used to quantify the number of distinguishable place fields on the maze. The rate maps were generated using an algorithm described by the following equations.

The Gaussian kernel used is given by:

$$g(x) = \exp\left(\frac{-x^2}{2}\right)$$

The algorithm for calculating firing rate is then given by:

$$\lambda(x) = \sum_{i=1}^{n} g\left(\frac{s_{i-x}}{h}\right) \Big/ \int_0^T g\left(\frac{y(t) - x}{h}\right) dt$$

where $S_i$ represents the positions of every recorded spike, *x* is the centre of the current bin, the period [0 T] is the recording session time period, *y(t)* is the position of the rat at time *t*, and *h* is a smoothing factor, which was set to 2.5 cm. Bins in which the rat did not explore within 5 cm of the centre were regarded as having never being visited.

## Place cell firing in the Centre Goal Box

If rats were aware that both Routes 2 and 3 led to the same goal box we would expect place cells to represent this location similarly regardless of the route taken there. Furthermore, we would expect higher correlations between the place cell activity in the Centre Goal Box when accessed via Route 2 and Route 3, than between activity in other pairs of boxes (e.g. the Left Box accessed via Route 1 and the Centre Goal Box when accessed by Route 2), and also that the activity in the Centre Goal Box when accessed by the two routes would more highly correlated than expected by chance.

To test this we generated a population vector, consisting of the firing rates of all place cells from all animals and sessions in the Centre Goal Box when the animals navigated using Route 2. We calculated the Spearman's ranked correlation between this vector and the population vector for the Centre Goal Box when animals navigated using Route 3. Lastly we calculated the ranked correlation between these two vectors and vectors for the Left and Right Goal Boxes; each of these boxes was represented by only one vector as there was only one possible route to each. This analysis provides six correlation values representing the comparison of the goal box activity at the end of each route to every other one. Next we calculated the probability that each of these values could occur by chance.

To do this we calculated the ranked correlation between each of the population vectors described above with a novel vector composed of random firing rate values from each cell (in effect shuffling route identity, whilst maintaining the order of the cells) after removing those firing rates belonging to the original population vector. This process was repeated 10000 times and the probability of the original correlation value occurring by chance was then estimated as the percentile position of that value in the distribution of correlation values resulting from the shuffled correlations (using a kernel smoothed cumulative density function). This analysis tells us the probability of any

two population vectors being similar by chance or in other words the likelihood that all place cells would fire similarly in two goal boxes by chance.

## Differential place cell firing

To assess whether place fields in areas of the maze that were traversed on more than one route were modulated by the route and/or by the goal, we focussed on activity occurring in four segments of the maze: the start box and the initial corridor or central stem (which were each common to all four routes and all three goals) and the right and left stems (each common to two routes and two goals) (*Figure 1C*).

The analysis for each segment was conducted only for neurons that had been identified as place cells in the maze environment, and that were active in that segment (active being defined as a mean firing >1 Hz in the maze segment on at least one of the four (or two) possible routes when all of the individual trajectories along one route were combined).

For each place cell that was active in a given segment of the maze, the firing rate in that segment was calculated for each trial (total number of spikes in segment/time in segment), together with the average x- and y- coordinates occupied by the animal for each trial, and the average velocity of the animal in that segment (total distance travelled in segment/total time spent there) for each trial. We then assessed whether firing rate differed between trials on which the animal had taken the four different routes using three different methods. See *Figure 3—figure supplement 1* for an example of the parameters used in the following analyses.

Method one, reported in the main text, used a nonparametric ANCOVA described previously (*Quade, 1967*). Briefly, this test consists of replacing the dependent variable (DV) and covariates (COV) with ranked equivalents. A linear regression is then performed on these ranked covariates against the DV, ignoring the independent variable (IV). A one-way ANOVA is then performed on the unstandardized residuals resulting from this regression against the original IV. If this was found to be significantly modulated by the animal's route ($p<0.05$) after controlling for the effects of the covariates, we conducted planned post-hoc tests. These consisted of six pairwise comparisons of the estimated marginal means between the four routes. This method was carried out in Matlab using the functions *tiedrank* for ranking, *fitlm* for the linear regression, *anova1* for the one way ANOVA and *multcompare* for the post hoc tests.

Method two employed a similarly nonparametric form of the ANCOVA test. We conducted an ANCOVA on the firing rate data using the same DV, IV and COV as above. We then compared the $F$-statistic obtained from this test to a distribution of $F$-statistics obtained after randomly shuffling the DV and repeating the ANCOVA. We then computed a p-value by the following method:

$$p = \frac{\#F_{shuff} \geq F_{obs}}{k}$$

where p is the probability that the observed $F$-value could have been obtained by chance, $F_{shuff}$ is the distribution of $F$-values obtained by shuffling the DV, $F_{obs}$ is the $F$-value obtained when testing the unshuffled DV and $k$ is the number of shuffles conducted. For our tests we conducted 5000 shuffles.

Method three employed a Generalized Linear Model (GLM) approach instead of an ANCOVA. We conducted a linear GLM on the DV, IV and COV described above with the underlying distribution assumed to be Poisson (*Di Lorenzo and Victor, 2013*) and a log link function. If firing rate was found to be significantly modulated by the animal's route ($p<0.05$) after controlling for the effects of the covariates, we conducted planned post-hoc tests. These consisted of six Mann-Whitney U tests comparing firing rates for every possible combination of the four routes. This method was carried out in Matlab using the functions *fitglm* to build the GLM and *ranksum* for the post hoc tests.

## Ensemble analysis

As an additional means of assessing the route- versus goal-related place cell firing, we tested whether the animal's trajectory could be derived from the ensemble activity of recorded place cells. If place cells show route-dependent activity at the ensemble level, then we would expect an automated route determination method to be equally accurate for all four routes, as each would be discriminable using the ensemble. In contrast, if ensembles reflect goal anticipation, then route

determination should be significantly less accurate for the two central routes, as both lead to the same goal.

Ensemble analyses were conducted separately for two segments of the maze: the start box and central stem. The animals traversed each of these zones on all trials and the trajectories within them should have remained similar regardless of which route the animals were taking. For each session and for each of these maze segments we compared the firing of all place cells on an individual trajectory (i.e. Route 1 to the Left Goal Box) to the average firing of these cells across all trajectories to each goal. In this way, we compared population vectors for every route to four, average firing rate, population vectors. However, the single trajectory being compared was not included in the calculation of its average goal vector, removing the possibility that it influenced the outcome of the assessment. To assess the similarity of each route population vector to each of the four goal vectors we used a cosine distance or cosine similarity measure defined as:

$$CosSim = \frac{\sum_i x_i y_i}{\sqrt{\sum_i x_i^2}\sqrt{\sum_i y_i^2}} = \frac{\langle x, y \rangle}{\|x\|\|y\|}$$

This calculation gives a value bounded between 0 and 1 if x and y are non-negative (such as firing rates). Cosine similarity can be interpreted as the cosine of the angle between two vectors, or as an alternative to the Pearson correlation that is sensitive to shifts in group values (i.e. If x is shifted to x + 1, the cosine similarity between x and y changes).

We next calculated the proportion of these comparisons which resulted in a 'correct' match between route vector and its corresponding goal vector and the proportion of those which suggested a similarity to one of the other goal vectors (a match was taken as the goal vector resulting in the highest similarity score). This was repeated for every session included in the analysis. In order to calculate the probability of correct matches being made by chance, we also repeated the above process 10000 times using four goal vectors where the identity of the route (but not of the contributing neuron) were shuffled, therefore disrupting any relationship between firing rate and the animal's trajectory. The probability that the proportion of correct matches made in our unshuffled analysis was the result of chance was estimated by calculating the percentile position of our observed proportion of matches in the distribution observed in the shuffled data. We did this using a kernel smoothed cumulative density function - Matlab function *ksdensity.* An Epanechnikov kernel was employed as it is one of the most widely used, optimal filters and we set bandwidth to the 'default' mode for all distributions. A brief outline of this process can be seen in *Figure 7*. We reasoned that if the central routes were represented more similarly (due to goal location dependent firing at an ensemble level) then we would expect an above chance level of matches between trajectories along Route 2 and the goal vector calculated for Route 3 (and vice versa) or we may expect fewer correct matches for the central routes than for the outer ones.

## Histology

At the end of the experiment animals were given an overdose of pentobarbital intraperitoneally (Euthatal, Merial Animal Health Ltd., Essex, UK), and perfused with 0.9% saline solution followed by a 4% formalin solution. The brain was extracted and stored in 4% formalin for at least seven days prior to any histological analyses. The brains were sliced in 32 μm sections on a freezing microtome at −20°. These sections were stained with a 0.1% cresyl violet solution and the slice best representing the electrode track was then imaged using ImageJ software (ImageJ, NIH, Bethesda).

## Acknowledgements

RMG is now at the Institute of Behavioural Neuroscience, Department of Experimental Psychology, University College London, London, United Kingdom. Grants: This work was supported by a grant from the UK Biotechnology and Biological Sciences Research Council (BBSRC; BB/L000040/1) to PAD.

## Additional information

### Funding

| Funder | Grant reference number | Author |
| --- | --- | --- |
| Biotechnology and Biological Sciences Research Council | BB/L000040/1 | Paul A Dudchenko |

The funders had no role in study design, data collection and interpretation, or the decision to submit the work for publication.

### Author contributions

Roddy M Grieves, Emma R Wood, Paul A Dudchenko, Conception and design, Acquisition of data, Analysis and interpretation of data, Drafting or revising the article

### Author ORCIDs

Paul A Dudchenko [iD] http://orcid.org/0000-0002-1531-5713

### Ethics

Animal experimentation: This experiment complied with the national [Animals (Scientific Procedures) Act, 1986, United 372 Kingdom] and international [European Communities Council Directive of November 24, 1986 (86/609/EEC)] legislation governing the maintenance of laboratory animals and their use in scientific experiments.

### Decision letter and Author response

Decision letter https://doi.org/10.7554/eLife.15986.sa1
Author response https://doi.org/10.7554/eLife.15986.sa2

## Additional files

### Supplementary files

• Supplementary file 1. Analysis of differential firing using three different statistical methods.

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
