## [Decision Letter]

Thank you for submitting your article "Do place cells encode goal locations or routes? Place fields on a maze with two routes to the same goal location" for consideration by *eLife*. Your article has been reviewed by two peer reviewers, and the evaluation has been overseen by a Reviewing Editor and Timothy Behrens as the Senior Editor. The reviewers have opted to remain anonymous.

The reviewers have discussed the reviews with one another and the Reviewing Editor has drafted this decision to help you prepare a revised submission.

Summary:

With this research the authors make an important contribution to our understanding of hippocampal processing by determining whether an animal's plan of action or its intended goal influences the activity of hippocampal neurons. The authors utilize a novel maze comprising multiple routes with overlapping goal locations to discern the ontology of "splitter" place cells, spatially selective neurons that exhibit differential firing rates according to the rat's intention. They find that a remarkably high number of neurons modulate their activity in anticipation of the route that the rat will take, demonstrating that the activity of the hippocampus reflects how the animal will accomplish its task and not just the intended goal. Though the experimental design is strong and the data are compelling, I have some concerns about the appropriateness of some of the analyses.

Essential revisions:

1) The authors correctly identify that the trajectory a rat takes through a region of interest will interact with the firing rate, as the trajectory determines whether the rat is aligned to the spatial center of the place field. However, rodents can tend to veer dramatically to one side corresponding to the intended destination, so this approach would not fairly address the possibility that the differential firing rate results from the rat incidentally veering into a stationary but off-center place field only on certain route trajectories. An ANCOVA is only appropriate if the covariates are not dependent on the independent variable. The authors should include whether they determined if the trajectory (reduced here to the mean x- and y-position) interacts with the route type for each ANCOVA.

On a related note, it is unclear how well the mean coordinate captures the idiosyncratic path through the region of interest. Even if each path is linear (at its most simple) the mean is still approximating two points, and each trajectory line could be crossing over others. The average coordinate is a good estimate if it can somehow be shown that each trajectory is being accurately and distinctly characterized. A visualization of an example showing this relationship would be a necessary component of reconciling this concern.

Because the distribution of firing rates across different passes through a cell's place field tends to be over-dispersed and not normal (Fenton and Muller, 1998), the authors should indicate whether the assumption of normality (or the normal approximation) was justified. Under-sampling of the distribution of each route (11-trial blocks) coupled with skewed distributions could lead to spuriously identified splitter cells.

The authors could consider using alternative methods for determining whether the firing rates are different between routes if vetting the normal statistics proves intractable. By using a bootstrapped Monte Carlo method to estimate the likelihood of finding the difference in firing rate distributions by chance, the issues with normality can be avoided (e.g. Prerau et al. 2014). Alternatively, or in addition, a GLM approach with continuous positional predictors in addition to categorical route predictors might be a fruitful framework to address all three concerns above.

2) The Pearson correlation was used to determine the similarity of a single trial ensemble vector to the trial-average route template, which acted to both estimate the reliability of coding a route and determine the discriminability of one route from another. The Pearson correlation assumes normality among the firing rates within the vector, and this is not a valid assumption since the rates come from different neurons. Alternatively, the cosine distance between the vectors could be used, which makes no assumption about the relationship among the firing rates of the cells.

3) One key question here is the definition of "goal." Certainly the goal location is the same in routes 2 and 3, but the initial choice in the central stem requires different turning directions and distinguishing subgoals. Quite different stories start with "once upon a time" and end with "and they lived happily ever after." Choices in the middle of behavioral episodes matter. Routes 2 and 3 were also confused most often (Figure 3), suggesting that representations of the different paths overlapped initially, and had to be separated, which is also consistent with the overrepresentation of the start of the maze. This hypothesis makes the strong prediction that early on, when confusion was likely, the number of trajectory coding neurons was likely low, and the authors could test this idea by testing the degree of trajectory discrimination by the population over the course of learning in each block.

4) Several studies suggest that CA1 neuronal activity distinguishes identical spatial trajectories and goals when nonspatial features are needed to select discriminative responses. Kennedy et al. (2009) showed that place field activity distinguished deprivation states during identical trajectories toward the same goal location. Ferbinteanu et al. (2011) found that place fields distinguished identical trajectories toward the same goal location depending on whether the rat used a place or cue approach strategy. Allen et al. (2016) showed differential activity as rats discriminated odors in particular sequences while remaining in a single place – the odor port. The real challenge is to define how animals parse the world. Taking different routes that start and end in the same place can have as few as one common goal, but as many different memories as choice points.

---

## [Author Response]

*Essential revisions:*

*1) The authors correctly identify that the trajectory a rat takes through a region of interest will interact with the firing rate, as the trajectory determines whether the rat is aligned to the spatial center of the place field. However, rodents can tend to veer dramatically to one side corresponding to the intended destination, so this approach would not fairly address the possibility that the differential firing rate results from the rat incidentally veering into a stationary but off-center place field only on certain route trajectories. An ANCOVA is only appropriate if the covariates are not dependent on the independent variable. The authors should include whether they determined if the trajectory (reduced here to the mean x- and y-position) interacts with the route type for each ANCOVA.*

*On a related note, it is unclear how well the mean coordinate captures the idiosyncratic path through the region of interest. Even if each path is linear (at its most simple) the mean is still approximating two points, and each trajectory line could be crossing over others. The average coordinate is a good estimate if it can somehow be shown that each trajectory is being accurately and distinctly characterized. A visualization of an example showing this relationship would be a necessary component of reconciling this concern.*

*Because the distribution of firing rates across different passes through a cell's place field tends to be over-dispersed and not normal (Fenton and Muller, 1998), the authors should indicate whether the assumption of normality (or the normal approximation) was justified. Under-sampling of the distribution of each route (11-trial blocks) coupled with skewed distributions could lead to spuriously identified splitter cells.*

The authors could consider using alternative methods for determining whether the firing rates are different between routes if vetting the normal statistics proves intractable. By using a bootstrapped Monte Carlo method to estimate the likelihood of finding the difference in firing rate distributions by chance, the issues with normality can be avoided (e.g. Prerau et al. 2014). Alternatively, or in addition, a GLM approach with continuous positional predictors in addition to categorical route predictors might be a fruitful framework to address all three concerns above.

In the revised document, we analysed the place cell firing as a function of the four possible routes the rat could take to the goal in three ways: with a non-parametric, ranked analysis of covariance, a non-parametric permutation analysis of covariance, and a generalised linear model (see subsections “Differential place cell activity throughout the maze” and “Differential place cell firing”, and Tables 1-4). All three analyses yield the same result: almost all cells with differential firing on the maze encode routes, as opposed to goals. We also provide a new figure (Figure 3—figure supplement 1), which illustrates the overlapping paths taken during a representative recording session.

2) The Pearson correlation was used to determine the similarity of a single trial ensemble vector to the trial-average route template, which acted to both estimate the reliability of coding a route and determine the discriminability of one route from another. The Pearson correlation assumes normality among the firing rates within the vector, and this is not a valid assumption since the rates come from different neurons. Alternatively, the cosine distance between the vectors could be used, which makes no assumption about the relationship among the firing rates of the cells.

As suggested by the reviewers, we have recomputed the ensemble analyses using the cosine difference or cosine similarity measure (subsection “Ensemble Analysis”). Both yield the same pattern of results as observed with the Pearson correlations (Results).

3) One key question here is the definition of "goal." Certainly the goal location is the same in routes 2 and 3, but the initial choice in the central stem requires different turning directions and distinguishing subgoals. Quite different stories start with "once upon a time" and end with "and they lived happily ever after." Choices in the middle of behavioral episodes matter. Routes 2 and 3 were also confused most often (Figure 3), suggesting that representations of the different paths overlapped initially, and had to be separated, which is also consistent with the overrepresentation of the start of the maze. This hypothesis makes the strong prediction that early on, when confusion was likely, the number of trajectory coding neurons was likely low, and the authors could test this idea by testing the degree of trajectory discrimination by the population over the course of learning in each block.

The reviewers raise an interesting question on the development of differential firing. In the revised version, we analysed ensembles of cells within blocks of training trials. No significant difference in the incidence of differential firing was observed within blocks (subsection “Ensemble activity suggests route-dependent firing, not goal location-dependent firing”; Figure 7—figure supplement 1). It may be, however, that this is due to the well-trained nature of the current task. In a separate study (Stevenson et al., in preparation), we have observed a lower incidence of differential during initial acquisition of a similar task.

4) Several studies suggest that CA1 neuronal activity distinguishes identical spatial trajectories and goals when nonspatial features are needed to select discriminative responses. Kennedy et al. (2009) showed that place field activity distinguished deprivation states during identical trajectories toward the same goal location. Ferbinteanu et al. (2011) found that place fields distinguished identical trajectories toward the same goal location depending on whether the rat used a place or cue approach strategy. Allen et al. (2016) showed differential activity as rats discriminated odors in particular sequences while remaining in a single place – the odor port. The real challenge is to define how animals parse the world. Taking different routes that start and end in the same place can have as few as one common goal, but as many different memories as choice points.

We fully agree with the reviewers that place cells “parse” the world in different ways, depending on the contingencies they experience. In the revised Discussion, we now consider additional findings of differential place cell firing based on motivation or task demands.